# Comparative Transcriptome Analysis of Bovine, Porcine, and Sheep Muscle Using Interpretable Machine Learning Models

**DOI:** 10.3390/ani14202947

**Published:** 2024-10-12

**Authors:** Yaqiang Guo, Shuai Li, Rigela Na, Lili Guo, Chenxi Huo, Lin Zhu, Caixia Shi, Risu Na, Mingjuan Gu, Wenguang Zhang

**Affiliations:** 1College of Animal Science, Inner Mongolia Agricultural University, Hohhot 010010, China; gggyaqiang@163.com (Y.G.); lishuai@emails.imau.edu.cn (S.L.); 17647621083@163.com (R.N.); 13474912747@163.com (L.G.); 15660097986@163.com (C.H.); zhulinynacxhs@163.com (L.Z.); shicx98@163.com (C.S.); narisu@swu.edu.cn (R.N.); 2Inner Mongolia Engineering Research Center of Genomic Big Data for Agriculture, Hohhot 010010, China

**Keywords:** muscle growth and development, machine learning, SHAP, key genes, comparative transcriptomics

## Abstract

**Simple Summary:**

This study aimed to conduct a comprehensive analysis of the factors involved in muscle growth and development, along with their associated biological functions and pathways, in *cattle*, *pigs,* and *sheep*. This was achieved by integrating an interpretable machine learning model with a comparative transcriptomic approach. The findings revealed that the elements closely linked to muscle growth and development across these three species exhibited high repeatability. Specifically, we identified 8 unique factors in *cattle*, 2 in *pigs*, and 1 in *sheep*, thereby providing novel insights into the mechanisms underlying muscle growth and development within these species.

**Abstract:**

The growth and development of muscle tissue play a pivotal role in the economic value and quality of meat in agricultural animals, garnering close attention from breeders and researchers. The quality and palatability of muscle tissue directly determine the market competitiveness of meat products and the satisfaction of consumers. Therefore, a profound understanding and management of muscle growth is essential for enhancing the overall economic efficiency and product quality of the meat industry. Despite this, systematic research on muscle development-related genes across different species still needs to be improved. This study addresses this gap through extensive cross-species muscle transcriptome analysis, combined with interpretable machine learning models. Utilizing a comprehensive dataset of 275 publicly available transcriptomes derived from porcine, bovine, and ovine muscle tissues, encompassing samples from ten distinct muscle types such as the semimembranosus and longissimus dorsi, this study analyzes 113 porcine (*n* = 113), 94 bovine (*n* = 94), and 68 ovine (*n* = 68) specimens. We employed nine machine learning models, such as Support Vector Classifier (SVC) and Support Vector Machine (SVM). Applying the SHapley Additive exPlanations (SHAP) method, we analyzed the muscle transcriptome data of *cattle*, *pigs*, and *sheep*. The optimal model, adaptive boosting (AdaBoost), identified key genes potentially influencing muscle growth and development across the three species, termed SHAP genes. Among these, 41 genes (including *NANOG*, *ADAMTS8*, *LHX3*, *and TLR9*) were consistently expressed in all three species, designated as homologous genes. Specific candidate genes for *cattle* included *SLC47A1*, *IGSF1*, *IRF4*, *EIF3F*, *CGAS*, *ZSWIM9*, *RROB1*, and *ABHD18*; for *pigs*, *DRP2* and *COL12A1*; and for *sheep*, only *COL10A1*. Through the analysis of SHAP genes utilizing Kyoto Encyclopedia of Genes and Genomes (KEGG) pathways, relevant pathways such as ether lipid metabolism, cortisol synthesis and secretion, and calcium signaling pathways have been identified, revealing their pivotal roles in muscle growth and development.

## 1. Introduction

*Cattle*, swine, and *sheep* are the primary meat sources globally, and their muscle tissues hold significant economic and nutritional value [1]. Among these, bovine and ovine meat boast high and high-quality protein content, whereas porcine meat tends to be higher in fat [2]. The growth and development of muscle tissue profoundly influence the quality of meat from these three agricultural animals [3]; however, the molecular mechanisms underlying these processes remain inadequately elucidated across species [4]. While studies have focused on the genes related to muscle development in individual species [5], significant gaps in knowledge exist regarding the systematic differences in gene expression during muscle growth and development among *cattle*, *pigs*, and *sheep* and how these differences impact the growth and quality of muscle tissue [6,7]. This research aims to systematically identify and compare the expression patterns of key genes in the muscle tissues of *cattle*, *pigs*, and *sheep* through extensive cross-species muscle transcriptome analysis, coupled with interpretable machine learning models, thereby providing a scientific foundation for optimizing breeding management and enhancing meat quality.

Currently, transcriptomics has provided comprehensive insights into the intricate mechanisms underlying muscle growth and development from various perspectives. Chunbo Cai et al. utilized RNA-Seq technology to investigate the transcriptome variances in the longest thoracic muscle of nine *horse-bodied pig* samples and nine *large white pig* samples. Their analysis identified a total of 3487 differentially expressed genes (DEGs), which serve as a valuable resource for studying the mechanisms and genes influencing skeletal muscle growth rate, meat quality, and energy metabolism [8]. Shenhe Liu et al. [9] conducted an RNA-Seq analysis to identify 200 significantly DEGs—81 down-regulated and 119 up-regulated—in heat-tolerant and heat-intolerant *Chinese Holstein cattle* (48 samples in total). Furthermore, they found that 14 of these candidate factors related to muscle development are involved in Protein–Protein Interaction (PPI) networks. The significance of DEGs in potential molecular mechanisms governing muscle growth and development was further emphasized by QTL and functional enrichment analyses in a meta-analysis of the open transcriptome data of six *sheep* by Seyedeh Fatemeh Hosseini et al. [10]. It is evident that while most studies focus on differential expression analysis using small sample sizes and single species, they often overlook the crucial aspect of complex gene interactions.

In recent years, the rapid advancement of high-performance computing and machine learning technologies has led to the extensive application of artificial intelligence in genomics and animal breeding [11]. By harnessing AI technologies, researchers have been able to process and analyze vast genomic data more efficiently, thereby illuminating complex biological phenomena and providing novel strategies and solutions for animal breeding [12]. These technologies not only improve genetic selection and enhance production efficiency but also expedite research into economically significant traits such as disease resistance, meat quality, and reproductive performance [13]. Polina Mamoshina et al. [14] utilized various machine learning model algorithms to compare the transcriptomic characteristics of “old” and “young” *human* muscles, predicting sample age. Their investigation employed feature importance analysis to pinpoint pivotal genes correlated with the senescence of muscle tissue. The findings elucidate that augmented cytoplasmic calcium concentrations, PPAR signaling, and the enhancement of neurotransmitter recycling serve as fundamental signaling cascades in the aging process of muscle. Furthermore, the engagement of the IGF-1R and PI3K-Akt-mTOR signaling axes is also associated with the age-related changes in muscle. Sana Farhadi et al. [15] integrated three publicly available *sheep* tail transcriptome datasets, identifying pivotal genes through differential analysis and validating these hub genes via machine learning algorithms. While these methods delved into the biological information of muscle transcripts from both single and multi-model perspectives, they fell short of dissecting the opaque, black-box models, leaving the rationale behind the model’s decisions shrouded in uncertainty.

This study presents an innovative, comprehensive approach, integrating various machine learning model algorithms, SHAP interpretation, Weighted Gene Co-expression Network Analysis (WGCNA), functional analysis, and PPI analysis for the first time, to conducting a comprehensive comparative study of the muscle transcriptomes of *cattle*, *pigs*, and *sheep*. Specifically, we have constructed and optimized multiple machine learning models to handle high-dimensional expression features, thereby accurately predicting the species category (*cattle*, *pigs*, *sheep*) of the input samples. In this process, we have introduced SHAP as an explanatory tool for the first time, quantifying the contribution of each feature to the prediction results and thereby delving into the internal logic of model decisions [16]. This innovative application not only enhances the interpretability of the models but also provides robust support in the selection of key genes. Compared to other studies, the superiority of our research is manifested in several aspects. Firstly, we have adopted a multi-index evaluation system to select the best model, ensuring the accuracy and reliability of the prediction results. Secondly, through SHAP interpretation, we have been able to visually display the ranking of contributions of the genes of the three species, thereby more precisely identifying the key genes affecting muscle growth and development. Finally, by integrating WGCNA, biological process, and protein interaction network analyses, we have further validated the biological significance of these key genes and revealed their potential as candidate markers for muscle growth and development. This study not only fills the gap in comparative analysis of the muscle transcriptomes of *cattle*, *pigs*, and *sheep* but also offers new perspectives and methodologies for animal breeding and muscle biology research by introducing advanced interpretable machine learning techniques. Our findings are expected to provide crucial clues for a deeper understanding of the mechanisms of muscle growth and development across different species and inject new vitality into the sustainable development of the livestock industry.

## 2. Materials and Methods

All data utilized in this study were derived from twelve publicly available biological projects within the Gene Expression Omnibus (GEO) database (https://www.ncbi.nlm.nih.gov/gds, accessed on 28 May 2024), specifically PRJNA305124, PRJNA488311, PRJNA659895, PRJNA447998, PRJNA416678, PRJNA1059045, PRJNA721166, PRJNA947491, PRJNA393239, PRJNA676109, PRJNA699585, and PRJNA676129. To ensure the diversity and representativeness of the data, we meticulously selected 275 transcript samples based on the following criteria: Firstly, we prioritized samples from genes exhibiting high expression levels in muscle tissues and associated with growth and development. Secondly, taking into account the physiological differences among various species, we selected 113 transcript samples from pig muscle, 94 from *cattle* muscle, and 68 from *sheep* muscle, encompassing ten different types of muscle tissues, including the *semimembranosus*, *longissimus dorsi*, *skeletal*, *gluteus maximus*, *gluteus medius*, *rectus femoris*, *supraspinatus*, *longissimus lumborum*, *adductor*, *psoas major*, and *longissimus thoracis muscles* (Appendix A). This selection strategy is designed to provide a comprehensive transcriptomic dataset to support our comparative study of muscle tissues across different species.

### 2.1. Data Retrieval and Quality Assurance

After downloading the SRA files from the GEO database, the raw sequencing files were converted to FASTQ format using SRA Toolkit (v3.12.0) [17]. Subsequently, the Seq2fun (v2.0.5) tool was utilized to analyze the muscle transcriptome FASTQ files, enabling automatic data quality control [18,19,20]. For cross-species transcriptome comparisons, we selected the mammalian database provided on the official website as the reference for sequence assembly and annotation. Following expression data analysis across different species, to mitigate discrepancies in library sizes across diverse samples and correct biases in library composition, we employed the Trimmed Mean of M-values (TMM) normalization method from the edgeR package (v3.34.0) [21]. The detailed procedure is as follows: We imported the raw count data and constructed a DGEList object that encapsulated gene expression counts and sample grouping information. To enhance the precision of our analysis, we utilized the filterByExpr function to eliminate genes with low expression levels across all samples. Subsequently, we calculated TMM normalization factors using the calcNormFactors function, thereby adjusting for differences in library sizes among samples. The final normalized Counts per Million (CPM) were obtained through the cpm function. To address batch effects inherent in the experimental setup, we utilized the removeBatchEffect function from the limma package (v3.52.1) [22]. Specifically, we created a design matrix that incorporated both the primary effects (experimental groups) and the batch effects (experimental batches). By applying the removeBatchEffect function, we effectively eliminated batch effects from the normalized data, preserving the effects of other factors.

### 2.2. Machine Learning and Model Interpretability

After data quality control, we used nine commonly utilized algorithms for machine learning classification models, including SVC, SVM, logistic regression (LR), decision tree (DT), k-nearest neighbors (KNN), plain Bayes (PB), adaptive boosting (AdaBoost), deep neural network (DNN), and recurrent neural network (RNN) to analyze the expression data (Table 1) in order to select the optimal model from them. During model training, 30 samples from each species were randomly selected as the test set, and the remaining samples were used as the initial training set. Due to the imbalance in the number of samples from different species, the Synthetic Minority Oversampling Technique (SMOTE) method was used to synthesize some of the data from *cattle* and *sheep* in the initial training set in order to converge the number of samples from each species in the training set [23,24]. Except for the plain Bayesian model, the other eight models were hyper-parameterized (Appendix A). Each model was evaluated by 5–fold cross-validation. To verify the classification effectiveness of the optimal model, precision, recall, F1–score, and the confusion matrix between the real and predicted species names classified in the test set were used for model evaluation (Table 1 and Figure 1).

Precision is employed to gauge the accuracy of the model when predicting positive classes. We define it as the proportion of samples that the model predicts to belong to a specific species (such as a particular biological category) that actually do belong to that species. Recall, on the other hand, is utilized to evaluate the model’s ability to identify the target species, representing the ratio of how many samples that truly belong to the species are correctly predicted as such. The F1–score is the harmonic mean of precision and recall, providing a balanced measure of the model’s accuracy and comprehensiveness. It is important to clarify that the F1 score does not result from simply adding or multiplying precision and recall; rather, it is calculated as the harmonic mean of these two metrics. This combination offers a single-valued measure that balances the model’s performance in terms of both accuracy and completeness, especially useful in cases of class imbalance. The confusion matrix presents the classification results in a matrix format, clearly illustrating the model’s predictive performance across different categories. Specifically, it depicts the relationship between the actual species and the predicted species, providing a detailed view of the model’s classification outcomes. By employing precision, recall, the F1 score, and the confusion matrix, we can comprehensively assess the classification effectiveness of the optimal model. These metrics reflect the model’s performance in species classification from various perspectives, aiding in the identification of strengths and weaknesses and offering guidance for subsequent optimization. In summary, these metrics are not combined through simple addition or multiplication but rather through their distinct roles in evaluating different aspects of the model’s performance. Precision and recall assess accuracy and completeness, respectively, while the F1–score provides a balanced view by considering both. The confusion matrix offers a comprehensive visualization of the classification results, helping to understand the model’s predictive capabilities across different categories.

SHAP values are a powerful tool for interpreting the predictive outcomes of machine learning models by quantifying the impact of each feature on the model’s predictions. In this study, we utilized SHAP values to evaluate the influence of genes on predicting the development and functionality of muscle tissue. SHAP values range from negative to positive values, where a positive SHAP value indicates a feature’s (gene’s) contribution to increasing the model’s output. The magnitude of the SHAP value reflects the strength of this contribution; a SHAP value greater than zero signifies that the gene positively impacts the model’s prediction, suggesting a crucial role in facilitating the development and functionality of muscle tissue.To clarify the significance of SHAP values, we performed a series of statistical tests to assess their reliability and significance. These tests included examining the distribution of SHAP values and their consistency across different models and datasets. The most positive SHAP values, which indicate the strongest positive influence on the model’s output, were determined based on the distribution of SHAP values for each gene across all samples. It is important to note that while the magnitude of SHAP values can be influenced by the input data, the relative importance of genes as indicated by their SHAP values remains a robust measure of their biological relevance within muscle tissue.For comparative analysis, we standardized the SHAP values across different models to ensure a fair comparison. This was achieved by normalizing the SHAP values to have a mean of zero and a standard deviation of one, allowing us to directly compare the importance of genes between different models and species. After model comparison, the most interpretable model was further analyzed using SHAP, and the top 100 generic gene IDs for *pigs*, *cattle*, and *sheep* were listed based on the magnitude of their SHAP values. Since only the top 75 of these 100 generic gene IDs across the three species had SHAP values significantly greater than zero, we focused on these top 75 genes for subsequent analyses (Appendix A).

The process was conducted using Python (v3.10) [25]. Pandas (v1.5.3) was utilized for data processing and analysis [26], while Scikit-learn (v1.2.2) provided a range of machine learning algorithms and tools, including model selection and performance evaluation metrics [27]. Charting and graphing were performed using Matplotlib (v3.7.1) [28], with Seaborn (v0.12.2) offering a more advanced plotting interface and esthetically pleasing default styles [29]. Imbalanced-learn was employed to address imbalanced datasets (v0.10.1) [30], and TensorFlow (v2.13.0), along with its Keras module, was used to construct and train deep learning models [31]. Additionally, Scipy (v1.11.3) contains statistical functions and probability distributions for generating random numbers and conducting statistical analysis [32].

### 2.3. Determination of SHAP Genes in Cattle, Pigs, and Sheep

Comparing the 75 universal gene IDs retained for each species with the mammalian database and filtering out the unannotated genes, we identified 49, 49, and 48 genes for *cattle*, *pigs*, and *sheep*, respectively (Appendix A). These genes are referred to as SHAP genes. Among these SHAP genes, those expressed in only one species are referred to as SHAP-specific genes, while those expressed in all three species are referred to as homologous genes. Comparative analysis of SHAP genes from different species was conducted using VennDiagram (v1.5.0) [33]. The average expression of homologous and species-specific SHAP genes was calculated using Excel. In order to investigate the relationships of homologous genes among the three species and the correlation between species-specific genes, we conducted a comprehensive correlation analysis. This analysis aimed to uncover the co-expression patterns and potential functional connections among genes. Utilizing libraries such as Pandas in Python, we first removed non-gene-related columns from the gene expression dataset, such as sample identifiers. Subsequently, we computed the correlation matrix for the remaining gene expression data. This matrix quantitatively represents the pairwise correlations among genes, with each cell in the matrix corresponding to the correlation coefficient between two genes. The correlation coefficients range from −1 to 1; values approaching 1 indicate a strong positive correlation, values nearing −1 signify a strong negative correlation, and values close to 0 suggest a lack of correlation. Subsequently, KEGG and Gene Ontology Enrichment Analysis (GO) of homologous genes for generic species were performed using the Expressanalyst online website (https://www.expressanalyst.ca/ModuleView.xhtml, accessed on 12 August 2024). The resulting csv files were obtained and visualized using the ggplot2 (v3.4.2) [34].

### 2.4. Cattle, Pigs, and Sheep Muscle Transcriptome WGCNA

To investigate co-expression patterns in gene expression data, we utilized the WGCNA method. In order to ensure the quality of the data, we employed the goodSamplesGenes function to evaluate both samples and genes and subsequently removed any non-compliant genes and samples. Following this, we constructed gene co-expression networks using power values as parameters for the weighting function, based on network topology analysis. The weighted co-expression network of genes was then built using the blockwiseModules function. WGCNA (v1.7.2) was utilized for weighted gene co-expression network analysis [35], providing functions for constructing networks, identifying modules, calculating gene correlations, etc. Additionally, we used Dplyr (v1.1.3) for data manipulation and processing. Tidyverse (v2.0.0) simplified the process of data processing and visualization, while ColorRamps (v2.3) was employed for generating color gradients and palettes. Lastly, we utilized DynamicTreeCut (v1.6.3) for dynamic tree clipping, which aids in performing dendrograms in WGCNA for clipping and module identification [36,37,38].

### 2.5. PPI Analysis

The Search Tool for the Retrieval of Interacting Genes/Proteins (STRING) is a bioinformatics database and tool primarily used for predicting and analyzing (https://cn.string-db.org/, accessed on 15 August 2024) [39]. In order to validate the biological functions of SHAP genes across different species, we conducted PPI analysis for the SHAP genes of each species on the STRING website. Genes that did not form networks were excluded, while the remaining ones were retained for visualization purposes.

## 3. Results

In this study, our aim was to investigate the genes and their mechanisms that are most relevant to muscle growth and development in *cattle*, *pigs*, and *sheep* by constructing interpretable machine learning models. Firstly, we analyzed the optimal AdaBoost models trained on muscle transcriptome expression data from the three species for interpretability using SHAP. We identified possible key generic gene IDs by ranking the SHAP values of the genes. Next, we compared these gene IDs with the gene names in the species database to identify potential key genes in different species. Subsequently, we conducted an analysis of homologous and specific genes to compare differences between species. Finally, we performed WGANA analysis and protein PPI network analysis to gain insight into the biology of these key genes.

### 3.1. Data Quality Control

Given that the data originated from multiple biological projects, we conducted principal component analysis on the filtered data and on the data after batch effects removal and normalization, respectively (Figure 1a,b). The processed data after batch effects removal and normalization exhibited significantly improved quality compared to the unprocessed data. It was able to more clearly differentiate the muscle transcriptome samples from *cattle*, *pigs*, and *sheep*. Subsequently, the 16,946 universal gene IDs were divided into 10 bins at every 10% interval after batch effect removal and normalization (Appendix A). The average expression of all genes in each 10% bin for *cattle*, *pigs*, and *sheep* samples was then calculated (Figure 1c). After calculating the average expression levels, we take the logarithm base 2 of these values(Similarly, the calculation for Figure 3b). From an evolutionary perspective, differences in the expression levels of muscle tissue genes among the three species were observed. For most genes, expression levels were higher in *pigs* than in *cattle* and higher in *cattle* than in *sheep*. Although these differences were not statistically significant, they reflect a certain degree of conservatism in the evolutionary process of these three species.

### 3.2. Evaluation of the Classification Performance of Nine Machine Learning Models on Samples of Cattle, Pigs, and Sheep

In order to achieve the optimal classification model, we conducted a comprehensive evaluation of the classification effectiveness of nine different machine learning models. Initially, precision, recall, and F1 score were utilized to visually represent the classification effectiveness of each model (Table 1). The findings indicate that the recurrent neural network exhibited the lowest F1 score, whereas the integrated model adaptive enhancement demonstrated an impressive F1 score of 0.99 and a classification accuracy of 0.99.

**Table 1 animals-14-02947-t001:** Precision, recall, F1–scores, and accuracy for nine models.

Module	Precision	Recall	F1–Score	Accuracy
SVC	0.89	0.89	0.89	0.89
SVM	0.78	0.78	0.78	0.78
DNN	0.88	0.83	0.87	0.84
RNN	0.76	0.72	0.73	0.74
LR	0.75	0.75	0.75	0.73
DT	0.89	0.87	0.89	0.89
KNN	0.90	0.90	0.90	0.9
NB	0.94	0.93	0.94	0.93
AdaBoost	0.99	0.99	0.99	0.99

To thoroughly evaluate the classification performance of the model, this study conducted a comparison between the predicted results and real species names. It was observed that the RNN exhibited lower F1 scores and accuracy compared to other models due to its prediction of all 30 bovine muscle transcript samples in the test set as porcine, as well as its incorrect prediction of nearly half of the *sheep* muscle transcript samples as either bovine or porcine (Figure 2). In contrast, the AdaBoost model only misclassified one out of 30 *sheep* muscle transcript samples as a *cattle* muscle transcript sample. These findings indicate that AdaBoost demonstrated superior classification results among the nine models evaluated in this study, positioning it as the most suitable model for our purposes.

### 3.3. Interpretability of Black-Box Models and SHAP Gene Selection and Analysis

After determining the optimal model for adaptive enhancement, this study utilized SHAP to analyze its interpretability. A comparison of the three species revealed that 41 homologous genes were co-expressed in all three species, while 8, 2, and 1 gene(s) were specifically expressed in *cattle*, *pigs*, and *sheep*, respectively (Figure 3a). Further statistical analysis of the expression of the 41 homologous genes in each species indicated that the expression level of the *RPS6* gene was significantly higher than that of the other 40 homologous genes (Figure 3b). Subsequently, the intraspecific intergenic correlations of the 41 homologous genes in different species were analyzed (Figure 3c–e). The results demonstrated significant differences in intraspecific correlations among different species. For instance, in the *sheep* muscle transcriptome, it was observed that the *GAP43* gene exhibited a strong negative correlation with the *CENPO* gene, whereas only weak negative and positive correlations were evident between these two genes in muscle transcripts from *cattle* and *pigs*. Based on these findings, this study tentatively inferred that key genes influencing muscle growth in *cattle*, *pigs*, and *sheep* are expressed at varying levels within muscles across different species.

Subsequently, this study conducted an analysis of the specific SHAP genes present in the muscle transcripts of bovine, porcine, and *sheep*. It was observed that among the eight specific SHAP genes in *cattle*, the expression level of the *RROB1* gene was significantly higher than that of *SLC47A1*, *IGSF1*, *IRF4*, *EIF3F*, *CGAS*, *ZSWIM9*, and *ABHD18*. Additionally, the expression of these seven genes remained more stable across different samples (Figure 4a). Three specific SHAP genes were identified in *pigs* and *sheep*; although their overall expression levels were not high, the expression of these three genes varied significantly among different samples within each species (Figure 4b,c). Amongst the six SHAP genes specifically shared by *pigs* and *sheep*, it was noted that the *FOXS1* gene exhibited much higher expression levels in certain *sheep* muscle samples compared to others as well as in pig muscle samples. On the other hand, the remaining five genes showed more similar expressions between both species (Figure 4d). Further analysis on the relationships amongst species-specific SHAP genes revealed that these 11 species-specific SHAP genes from all three species displayed significantly more negative correlations than positive ones (Figure 4e).

### 3.4. KEGG and GO Analysis of SHAP Homologous Genes

To validate the accuracy of the biomarker genes screened by the interpretable machine learning model, we conducted KEGG and GO analyses on homologous genes from 41 different species. The KEGG pathway analysis revealed related pathways such as ether lipid metabolism, cortisol synthesis and secretion, and the calcium signaling pathway (Figure 5a). In the GO enrichment analysis, biological processes including innate immune response, lipid and lipid metabolic processes, and calcium ion transport were identified; cellular components such as endoplasmic reticulum membrane of the peroneal membrane and collagen-containing extracellular matrix were identified; while molecular functions such as tyrosine kinase activity of transmembrane receptor proteins, G-protein-coupled receptor activity, and ATP binding were also identified (Figure 5b). Based on these results, it is reasonable to believe that the genes screened by the interpretable modeling algorithm in this study provide valuable insights into musculogenesis, growth, and development across different species.

### 3.5. WGCNA

The WGCNA identified eight gene modules, which were represented by different colors in the network. The number of genes within each module ranged from 43 to 832 (Figure 6a,b). These module-specific genes showed correlations with *cattle*, *pigs*, and *sheep* species. Notably, the turquoise module containing 832 genes exhibited the highest correlation of -0.80 with pig muscle transcripts, while the red and blue modules had higher correlations for *cattle* and *sheep*, respectively (Figure 6c). Upon comparison with SHAP genes screened by machine learning, it was found that only four SHAP genes (RPS6, RPA2, DARS, and MED15) were recognized by all three modules.

### 3.6. PPI Analysis within the Species of Cattle, Pig, and Sheep

By conducting PPI network analysis of SHAP genes from porcine, bovine, and *sheep* muscle transcriptome data, we have successfully constructed network maps containing core muscle developmental and functional proteins in the three species. The networks for bovine, porcine, and *sheep* contained 15, 12, and 8 key genes and their interactions, respectively. These findings demonstrate the regulatory networks involved in muscle development. It was observed that each animal’s regulatory networks presented unique topological and functional features during muscle development (Figure 7a–c). The porcine PPI network highlights several key proteins that play important roles in muscle growth and repair with a high degree of network concentration. In contrast, the *cattle* network revealed more protein interactions related to energy metabolism, suggesting a focus on the regulation of energy homeostasis within their muscle transcripts. On the other hand, the PPI network of *sheep* showed a pattern intermediate between that of *pigs* and *cattle*, indicating characteristics of both types of biology. Despite interspecies differences, common interactions within the PPI networks reveal conserved pathways for muscle development across all three animals. These results not only deepen our fundamental understanding of livestock muscle biology but also provide potential molecular targets for future genetic modification aimed at improving meat quality.

## 4. Discussion

Cross-species transcriptomic analysis represents a scientific approach that extracts pivotal information from diverse datasets, aiding in the identification of potential biological insights [40]. Currently, transcriptomic studies predominantly employ several methodologies, including the analysis of primary research data, integrative analysis of publicly available datasets, and a combination of both [41]. However, when confronted with the pronounced heterogeneity and limited sample sizes inherent in primary studies, models are susceptible to overfitting, resulting in the diminished validity of outcomes and reduced consistency of research findings [42]. Reckless amalgamation of data can also lead to significant errors. This study illustrates the limitations of such indiscriminate integration, revealing that transcriptomic data from the muscle tissues of *cattle*, *pigs*, and *sheep* sourced from over a dozen origins demonstrate poor consistency during unprocessed principal component analysis (Figure 1a). Thus, merely aggregating data from varying origins without appropriate processing may prevent effective differentiation between samples and hinder the identification of crucial biomarkers specific to particular species. In response to this challenge, we employed the limma package in R and utilized the TMM function in EdgeR to eliminate and normalize batch effects across datasets derived from different origins. Furthermore, we addressed missing values using the median interpolation method. While it cannot be asserted that these techniques are standard practice in all circumstances, our results indicate that after the application of these two methods, batch effects have been effectively corrected. Consequently, muscle transcriptomic expression samples from *pigs*, *cattle*, and *sheep* can now be distinctly differentiated (Figure 1b), suggesting that the processed and imputed data distribution closely resembles that of real-world data.

Following meticulous deliberation in this investigation, the AdaBoost model was ultimately selected as the machine learning paradigm. Due to the inherent particularity and intricacy of the data, there exists no universally applicable machine learning algorithm capable of accommodating diverse data types; hence, tailored machine learning models are imperative for distinct data categories [43,44]. This inquiry centers on the evaluation of nine widely utilized machine learning models, inclusive of two deep learning architectures. Upon thorough comparative analysis, AdaBoost emerged as the superior choice, distinguished by its elevated F1 score and exemplary classification efficacy. Despite their intrinsic complexity and enhanced capacity to accurately fit intricate datasets [45], the two neural network models exhibited inferior performance relative to the conventional tree and linear models in this context. Additionally, attempts to fine-tune their hyperparameters revealed a paradoxical decline in classification proficiency as the parameter complexity escalated. Consequently, this study posits the possibility of underfitting in the two deep learning models, attributable to both the intrinsic complexity of the data and the marginal heterogeneity among samples (Table 1, Figure 2c,d). In contrast, the AdaBoost algorithm demonstrates superior aptitude for addressing complex data classification challenges by amalgamating multiple weak classifiers to construct a robust classifier [46]. The constituent subclassifiers employ a CART decision tree, characterized by its simplistic binary classification structure. Considering the volumetric complexity of our dataset and the substantial sample size in this study, the training process was notably time-consuming; however, its most salient attribute lies in its reduced susceptibility to overfitting during the training phase, while the iterative process effectively amplifies margins [47].

Despite having identified the optimal machine learning model through comparative analyses, challenges arose when applying SHAP for interpretability. SHAP offers a variety of interpretation techniques, including Tree SHapley Additive exPlanations (tree SHAP) [48], Exact SHapley Additive exPlanations (Exact SHAP) [49],Kernel SHapley Additive exPlanations (Kernel SHAP) [50], and linear SHapley Additive exPlanations (linear SHAP) [51], each with distinct applicability and computational complexities. For instance, linear SHAP is confined to linear models, whereas Exact SHAP, Sampling SHAP, and Kernel SHAP, while capable of interpreting any model, impose significant computational burdens on complex data, demanding substantial computational resources.

In the present study, we opted for the tree SHAP method, renowned for its superior performance and modest computational resource requirements. However, as the selected model, AdaBoost, is an ensemble model, it cannot be straightforwardly interpreted using conventional tree model methods. Consequently, we enhanced tree SHAP by computing the SHAP values for each tree and then averaging these values. This refined approach adeptly consolidates the SHAP information from all trees within the ensemble model, thereby providing comprehensive elucidations for model predictions.

Given that this study is predicated upon a cross-species analysis of muscle transcriptomes, it is abundantly clear that conventional analytical methodologies fall short of our needs. The employment of such methods would substantially hinder the advancement of our research. Consequently, we have harnessed the Seq2fun mammalian reference database to scrutinize the muscle transcriptome data extracted from *cattle*, *pigs*, and *sheep*. To enable comparative analysis across these three species, expression matrices were constructed utilizing universal gene IDs.

During the screening process for SHAP genes, we retained records of generic gene IDs ranking in the top 100 SHAP values pertinent to each category of genes contributing to the model (bovine, porcine, and ovine). Given that only the top 75 ranked generic gene IDs for each species exhibited SHAP values exceeding 0, these genes were compared with their respective nomenclatures in the database. This exercise yielded 49 candidate genes potentially linked to muscle growth and development in both *cattle* and *pigs*; similarly, 48 candidate genes were identified in *sheep*. Remarkably, 41 genes were discovered to be expressed across all three species, which we have classified as homologous genes within this context. Intriguingly, some commonality was noted in the expression patterns of these homologous genes across differing species.

Specifically, the *RPS6* gene demonstrated elevated expression levels in all three species when compared to other genes. Established studies have identified the phosphorylation of *RPS6* as a determinant of muscle strength in mice, with *RPS6* frequently employed as a metric for assessing muscle anabolism in human skeletal muscle [52,53]. Subsequently, 11 genes expressed uniquely in the three species were identified. Among these, pig *DRP2* emerged as an anti-muscular dystrophy-associated protein in rats, effectively inhibiting muscular dystrophy and exhibiting a close association with normal skeletal muscle development in humans [54,55]. *COL12A1* bears significant relevance to the development of connective tissues and tendons, with structural variants potentially leading to congenital severe muscle weakness in humans [56]. Mutations within *COL10A1* in *sheep* have been linked to specific muscle disorders, including congenital muscular dystrophy [57]. The bovine *IGSF1* was highlighted in mouse skeletal muscle as a candidate gene possibly linked to muscle mass [58]. *EIF3F* was identified as a regulator of skeletal muscle mass, particularly during instances of muscle atrophy [59,60].

This series of auspicious findings unequivocally substantiates that our research endeavors are not merely speculative. The SHAP genes, along with the SHAP-specific genes derived from our screening utilizing seq2fun, machine learning, and SHAP methodologies, are paramount to understanding muscle growth and development.Although this study obtained several modules with high correlation through WGCNA, only four duplicates were identified using the SHAP method for the screened genes. Additionally, these four genes are homologous in the three species, except for *RPS6*. Previous discussions have linked phosphorylation of *RPA2* to spinal muscular atrophy, an autosomal recessive disorder resulting in paralysis and severe muscle atrophy [61]. *DARS* is significantly enriched by its expression in the pectoral muscle of chickens, where it plays a crucial role in muscle development and intramuscular fat deposition [62]. *MEED15* was identified as one of the eight central genes for lipid metabolism in adult *cattle* and exhibited high expression levels in muscle and adipose tissue [63]. These results do not imply that the use of machine learning combined with SHAP interpretation is incorrect. WGCNA and its algorithms solely consider the relationship between genes and samples or phenotypes from a mathematical perspective while ignoring gene-to-gene connections [64]. On the other hand, SHAP interpretation not only encompasses the advantages of WGCNA but also compensates for its shortcomings. For instance, the optimized tree SHAP visualizes the relationship between each gene and other genes by constructing a decision tree. By calculating possible combinations between genes, their contributions to different combinations, marginal contributions, and utilizing the cumulative nature of SHAP values; it ultimately calculates each gene’s contribution to classification models—which also represents all genes’ contribution to a certain species—thereby illustrating the model’s decision-making process. In order to comprehensively verify the biological functions of our screened candidate genes, this study conducted KEGG, GO, and PPI analyses for SHAP genes across different species. In contrast to previous studies, the majority of genes identified in this study are not directly involved in muscle growth and development. Instead, they indirectly impact muscle growth and development through the synthesis of intermediate metabolites or important synthesis factors, as well as the production of key bioactive enzymes. This presents novel ideas and directions for future research.

## 5. Conclusions

In summary, this study successfully identified genes closely associated with muscle growth and development in *cattle*, *pigs*, and *sheep* by integrating cross-species muscle transcriptomic data, employing advanced machine learning techniques, and utilizing interpretable models. Our preliminary analysis results suggest that these genes may play a pivotal role in the muscle growth and development processes across different species. These genes are not only of significant importance in breeding programs, particularly in optimizing muscle growth and enhancing meat quality, where the selection of animals with specific gene markers can be achieved, but they also hold potential applications in future therapeutic interventions, such as in the treatment of conditions like muscle atrophy. Through KEGG pathway analysis of genes closely related to muscle in the three species, we further identified relevant pathways such as ether lipid metabolism, cortisol synthesis and secretion, and calcium signaling pathways, validating our initial hypothesis. Although this study provides preliminary evidence for the potential roles of these genes in muscle growth, their specific biological functions require experimental validation. Future research should focus on functional verification experiments to further substantiate the reliability and sensitivity of these genes, thereby providing a more robust scientific foundation for practical applications.

## Figures and Tables

**Figure 1 animals-14-02947-f001:**
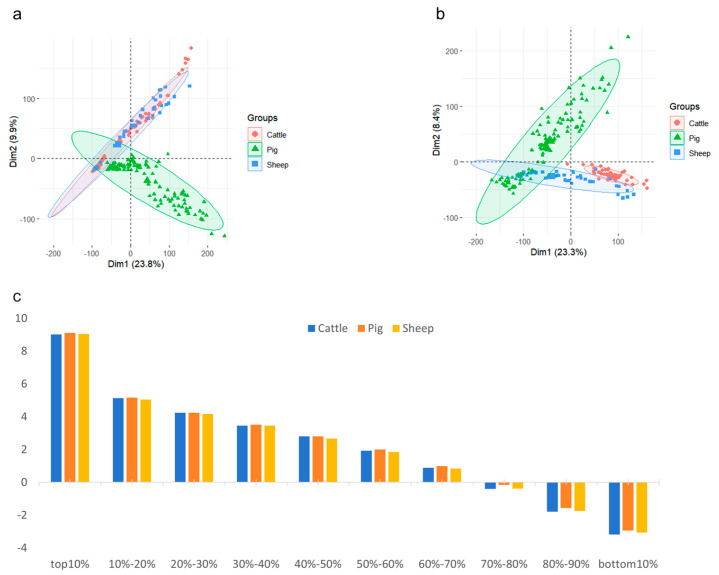
Preprocessing of muscle transcript sample expression data prior to model construction. (**a**) Principal component analysis plot used for filtering out low-expression genes. (**b**) Principal component analysis plot after the removal of low-expression genes, batch correction, and normalization. (**c**) Quality control of expression data and average gene expression analysis per 10% gradient.

**Figure 2 animals-14-02947-f002:**
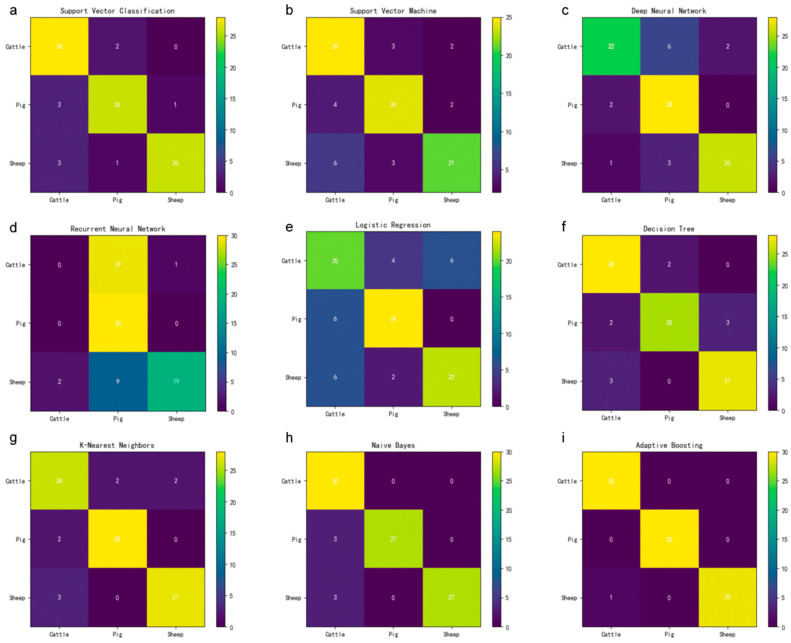
Evaluation of the classification performance of nine machine learning models on the test set. (**a**) Support Vector Classifier (SVC), (**b**) Support Vector Machine (SVM), (**c**) deep neural network (DNN), (**d**) recurrent neural network (RNN), (**e**) logistic regression (LR), (**f**) decision tree (DT), (**g**) k-nearest neighbors (KNN), (**h**) Naive Bayes (NB), (**i**) AdaBoost. In the confusion matrix, the modules where the species names correspond both horizontally and vertically represent the model’s accurate predictions of the actual conditions. The modules on either side of the diagonal indicate erroneous predictions of species names. Each species in the test set has a transcription sample size of thirty.

**Figure 3 animals-14-02947-f003:**
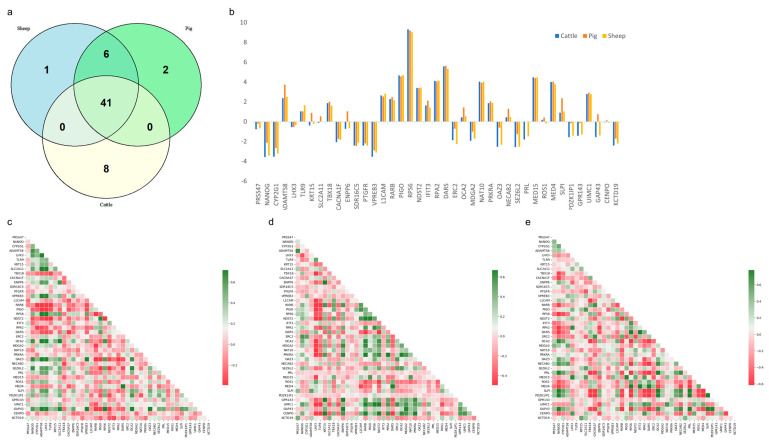
Analysis of SHAP genes in three species: *cattle*, pig, and *sheep*. (**a**) Venn diagram illustrating the distribution of SHAP genes among different species. (**b**) Expression levels of 41 homologous genes in the three species depicted on a bar graph, with gene names labeled on the horizontal axis and the corresponding expression levels on the vertical axis. (**c**) Correlation analysis among 41 genes in the bovine muscle transcriptome is presented, with gene names labeled on both the horizontal and vertical axes. (**d**) Correlation analysis among 41 genes in the porcine muscle transcriptome is shown. (**e**) Correlation analysis among 41 genes in the *sheep* muscle transcriptome is displayed.

**Figure 4 animals-14-02947-f004:**
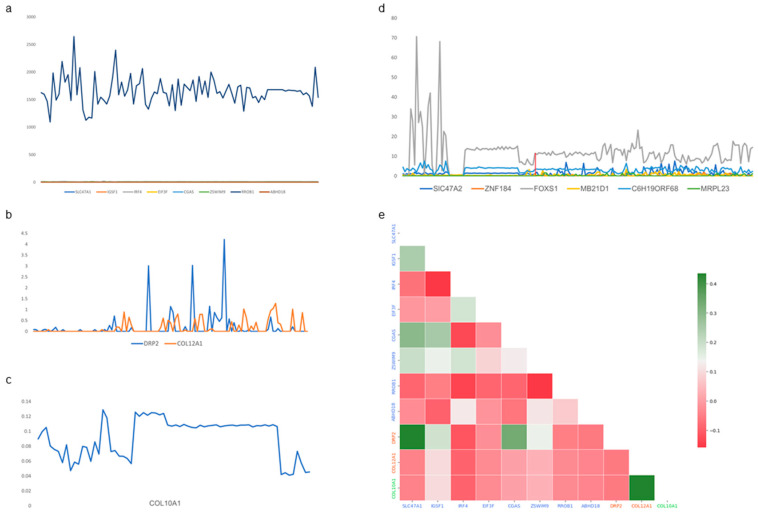
Intraspecies mean expression and interspecies correlation analysis of bovine, porcine, and *sheep*-specific SHAP genes. (**a**) Analysis of the average expression of bovine muscle-specific SHAP genes. (**b**) Analysis of the average expression of pig muscle-specific SHAP genes. (**c**) Analysis of the average expression of *sheep* muscle-specific SHAP genes. (**d**) Analysis of the average expression of SHAP genes specific to *pigs* and *sheep*. (**e**) Analysis of the correlation among SHAP genes across bovine, porcine, and *sheep* species.

**Figure 5 animals-14-02947-f005:**
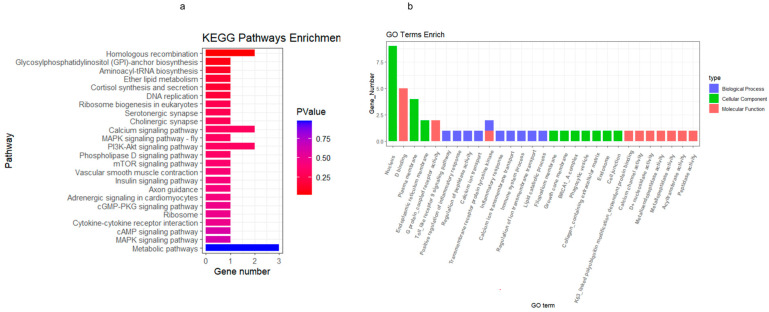
Presentation of the KEGG and GO analyses of the 41 homologous genes identified by the interpretable ML model SHAP across *cattle*, pig, and *sheep* species. Panel (**a**) shows the results of the KEGG enrichment analysis for the 41 homologous genes, while panel (**b**) displays the GO analysis findings for these genes.

**Figure 6 animals-14-02947-f006:**
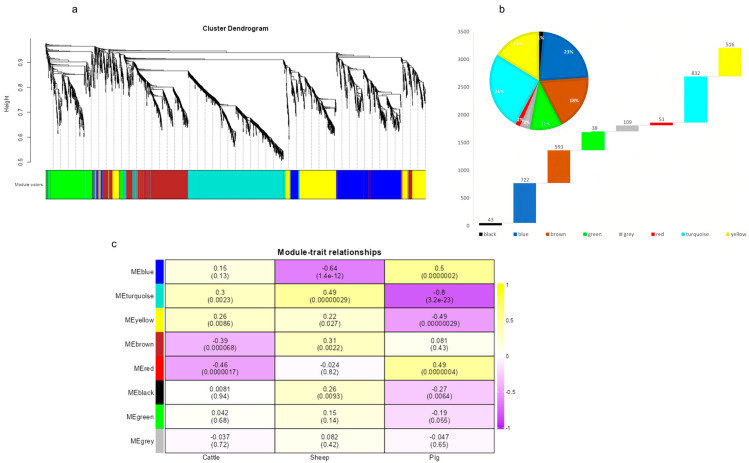
WGCNA was conducted among three species: *cattle*, *pigs*, and *sheep*. (**a**) Eight modules were retained after the removal of low-quality genes and samples. (**b**) The number of genes contained in each module and the percentage of the total number of genes. (**c**) A heatmap illustrating the correlation between the three species of *cattle*, *pigs*, and *sheep* with the eight characterized modules after the addition of species phenotypes.

**Figure 7 animals-14-02947-f007:**
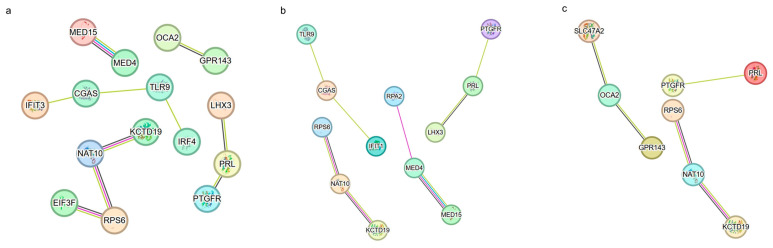
PPI network analysis of 49, 49, and 48 SHAP genes in muscle of *cattle*, pig, and *sheep*. (**a**) PPI network map of 15 key genes in 49 SHAP genes in *cattle*. (**b**) PPI network maps of 12 key genes in 49 SHAP genes in *pigs*. (**c**) PPI network map of 8 key genes in 48 SHAP genes of *sheep*.

## Data Availability

The data outlined in the paper, along with the code book and analytical code, can be accessed upon request.

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
