# Peer review of "Comparative Transcriptome Analysis of Bovine, Porcine, and Sheep Muscle Using Interpretable Machine Learning Models"

_animals, 2024, doi:10.3390/ani14202947_

Round 1

Reviewer 1 Report

Comments and Suggestions for Authors

The manuscript needs revision. Please refer to comments given in the text of reviewed attached file of the manuscript.

Author Response

Dear Editor and Reviewers,

We extend our heartfelt gratitude for your meticulous review and invaluable feedback regarding our manuscript titled "Comparative Transcriptome Analysis of Bovine, Porcine, and Sheep Muscle Using Interpretable Machine Learning Models." We hold each of your suggestions in high regard, and we have diligently considered and amended our work in accordance with your remarks. Below, we provide our detailed responses to the reviewers' comments:

In the revised article summary, we have meticulously delineated the work we have undertaken to address the identified gaps, briefly augmented our materials and methods, and incorporated specific conclusions within the detailed results. Subsequent to this, in the revisions of the introduction, materials and methods, as well as the conclusions, we have devoted considerable thought and diligent corrections to your queries and suggestions. Drawing upon the feedback of several reviewers, we have made substantial amendments to the article summary, introduction, materials and discussion, and conclusion sections. To facilitate your re-examination, we kindly request your renewed scrutiny of the revised manuscript, particularly regarding specific details. Additionally, for certain data, we have provided comprehensive supplementary information in the accompanying tables.

We sincerely thank you for your support and guidance throughout our work, and we eagerly await your further review. We hope that these modifications meet your expectations and facilitate the timely advancement of the paper’s publication process.  

With best regards!

Reviewer 2 Report

Comments and Suggestions for Authors

Comments on the Quality of English Language

They are included in the report.

Author Response

Dear Editor and Reviewers,

We extend our heartfelt gratitude for your meticulous review and invaluable feedback regarding our manuscript titled "Comparative Transcriptome Analysis of Bovine, Porcine, and Sheep Muscle Using Interpretable Machine Learning Models." We hold each of your suggestions in high regard, and we have diligently considered and amended our work in accordance with your remarks. Below, we provide our detailed responses to the reviewers' comments:

In response to the inquiries concerning the background and objectives in the abstract, we have made revisions per your recommendations, elucidating certain technical terms with their full names and definitions. Moreover, for some abstract terminology, readers are encouraged to contact any author via email for clarification. All genes mentioned in the manuscript have been abbreviated, with italicized annotations, and the complete names for the genes of each species are listed in Table S5.

In the introduction, we have rectified all grammatical errors and formatting issues while augmenting the text with our research objectives and questions.

In the materials and methods section, we have clarified the rationale behind sample distribution in the revised manuscript. Given that our data derive from publicly available datasets, it has proven challenging to ensure uniform sample sizes across the different species due to variations in data quantity and quality. We have included a description of the versions and standardization and normalization methods used for the various libraries. Regarding model evaluation, we have elaborated on the assessment methods mentioned and clarified our process for selecting SHAP genes. After determining the optimal classification model through five-fold cross-validation, we employed SHAP to elucidate the genes exerting the greatest influence on the muscle transcript samples of each species, arranging them in descending order based on SHAP values. All subsequent analyses are predicated upon this foundation. The reliability of genes selected through SHAP has been corroborated by numerous studies, a point we have addressed in the revised manuscript. The choice of these nine algorithms was motivated by their frequent utilization in classification tasks. We have also supplemented Table S4 with details regarding the hyperparameters for each model.

In the discussion and conclusion sections, we undertook significant revisions based on your recommendations and inquiries. In the initial version, we postulated associations between the genes identified through machine learning and SHAP values and the growth and development of the various species. We have now validated this through KEGG analysis. We acknowledge that our previous expression may have led to misunderstandings, which have now been rectified.

Once more, we thank the reviewers and the editor for their thorough evaluation and constructive suggestions. Your feedback has been instrumental in enhancing the quality of our manuscript. We eagerly await your review of our revised submission and hope that these amendments meet your expectations.

With best regards.

Reviewer 3 Report

Comments and Suggestions for Authors

The authors used bioinformatic tools to integrate large-scale cross-species muscle transcriptome data  and analyses with machine learning models.  Bovine, porcine, and ovine muscle transcriptome data were analysed to identify SHAP genes retrieved from the SHapley Additive exPlanations (SHAP) approach.

Among these 41 genes, NANOG, ADAMTS8, LHX3, and TLR9 were found to be expressed in all three species. Species specific candidate included for cattle SLC47A1, IGSF1, IRF4, EIF3F, CGAS, ZSWIM9, RROB1, ABHD18; for pigs: DRP2 and COL12A1;  and for sheep: COL10A1.

Their biological functions were explored using Kyoto Encyclopedia of Genes and Genomes (KEGG), Gene Ontology (GO), and Protein-Protein Interaction (PPI) methodologies.

Comments:

The manuscript does not contain line numbers. This creates difficulties to marke the respective text passages.

Simple summary

 The findings revealed that the regulators closely: please use another word for regulators. This can be misunderstood as you did search for regulatory variants. Please try to make clear the meaning of your results.

The same problem in the Abstract

Text: "Instead, this study proposes a comprehensive approach using multiple machine learning models for algorithmic comparison, SHAP for optimal black-box model interpretation, Weighted Gene Co-expression Network Analysis (WGCNA), functional analysis of muscle transcriptome profiles from different species, and PPI analysis. In summary, expression features are processed by multiple machine learning models to predict the specific species of pig, cattle, or sheep corresponding to the input sample.SHAP is a tool for interpreting machine learning models that facilitates understanding of modeling decisions by assigning the contribution of each feature to the predicted outcome [16]. The best model was evaluated using multiple metrics and then interpreted with SHAP to demonstrate the model decisions with gene SHAP values and rankings for the three species. This process identified key genes that may affect muscle growth and development in cattle, pigs and sheep. Subsequent analyses of WGCNA and biological processes as well as protein interaction networks demonstrated that these final screened key genes have potential as candidate markers for muscle growth and development. They are expected to provide new ideas and directions for studying muscle in these three species."

This text should be re-written. It becomes not clear which are the key parameteres and threshold to identify key genes from transcriptome data.

There can be usedmany different parameters to identify these genes. It looks that you did not just look on DEGs. Please explain in more detail.  

 Text" After data quality control, we used nine machine learning model algorithms including support vector classification(SVC), support vector machine(SVM), logistic regression(LR), decision tree(DT), k-nearest neighbor(KNN), plain Bayes(PB), adaptive boosting(AdaBoost), deep neural network(DNN), and recurrent neural network(RNN) to analyze the expression data (Table 1) in order to select the optimal model from them"

Please explain the optimal model. A schematic illustration on the different program steps and the thresholds used for inclusion/exclusion should be provided.

text"optimal model was analyzed for interpretability using SHAP, and the top 100 generic gene IDs for pigs, cattles, and sheep were printed based on the ranking of SHAP values"

why did you select 100 generic gene IDs

F1 score: can you explain what does this score mean

optimal classification model: how did you define this optimal model.

Legend of Figure 2: has to be improved. The meaning of this figure is not clear.

"In summary, through a combination of cross-species muscle transcriptomes, machine learning, and interpretable models, we identified genes that are strongly associated with muscle growth and development in cattle, pigs, and sheep.  " Please explain why these genes are associated with muscle growth and development in cattle, pigs, and sheep. Association analyses were not performed. Therefore, you looked for other parameters than association.

Comments on the Quality of English Language

No comments

Author Response

Dear Editor and Reviewers,

We extend our heartfelt gratitude for your meticulous review and invaluable feedback regarding our manuscript titled "Comparative Transcriptome Analysis of Bovine, Porcine, and Sheep Muscle Using Interpretable Machine Learning Models." We hold each of your suggestions in high regard, and we have diligently considered and amended our work in accordance with your remarks. Below, we provide our detailed responses to the reviewers' comments:

  1. Line numbers have been added, and the issues in the abstract and introduction have been rectified.  
  2. The description of the key gene selection process within the introduction has been refined, and a detailed explanation of the selection process is provided in the materials section.  
  3. Regarding the selection of the optimal model, we have elucidated the criteria in the materials section: a higher recall rate, precision, and F1 score indicate superior classification performance, thus identifying it as the optimal model.  
  4. Details on the hyperparameter tuning of the model are available (refer to Appendix 4), and specific machine learning code can be obtained by contacting the corresponding author.  
  5. In terms of the rationale for selecting 100 universal gene IDs, we have consulted a plethora of literature and found in practice that only a handful of genes contribute positively among a vast array. This phenomenon is reminiscent of the “80-20 principle”[1]in economics; in our biological realm, significant genes are often few. While this explanation may seem somewhat superficial, we aspire to articulate this complex issue in a more accessible manner.  

     6.The definition of the F1 score has been supplemented, along with a detailed description of                    Figure 2 provided in the materials section.  

We sincerely thank you for your support and guidance throughout our work, and we eagerly await your further review. We hope that these modifications meet your expectations and facilitate the timely advancement of the paper’s publication process.  

With best regards!

  1. The "80-20 Rule" (also known as the Pareto Principle) is a fundamental tenet in the realms of economics and management, germinating from the scholarly investigations of the Italian economist Vilfredo Pareto. This principle posits that, across a multitude of phenomena, approximately 80% of outcomes can be attributed to 20% of causes. This implies that, in numerous instances, a scant few factors wield substantial influence over the resultant effects, while the majority of factors exert a comparatively minor impact.

Round 2

Reviewer 2 Report

Comments and Suggestions for Authors

Animals-3227268-Peer-Review-Reportv2

Abstract

Line 10: Remove the word ‘both’ as it appears to be redundant.

Line 14-15: The tone needs to be readjusted to improve connections. Suggestion: Despite this, systematic research on muscle development-related genes across different species still needs to be improved.

Introduction

Lines 79-81: The word ‘ageing’ is used thrice; check the spelling.

Line 53: The authors stated that ‘Chunbo Cai et al. utilized….’. Kindly included the number corresponding to the order of references in square bracket as you have done in lines 57/58 (Shenhe Liu et al. [9]). Effect the same treatment for lines 75/76 and 82.

Lines 98-106: Split this sentence into 4. Suggestion: Compared to other studies, the superiority of our research is manifested in several aspects. Firstly, we have adopted a multi-index evaluation system to select the best model, ensuring the accuracy and reliability of the prediction results. Secondly, through SHAP interpretation, we have been able to visually display the ranking of contributions of the genes of the three species, thereby more precisely identifying the key genes affecting muscle growth and development. Finally, by integrating WGCNA, biological process, and protein interaction network analyses, we have further validated the biological significance of these key genes and revealed their potential as candidate markers for muscle growth and development.

Materials and Methods

Lines 124-126: Try and italicise the names of those muscles. Do same for other muscles throughout the text. Check ‘Ongissimus dorsi’, any muscle like that?

Author Response

Dear Esteemed Reviewer,

I extend my heartfelt gratitude for your most considerate and meticulous second review of my manuscript, as well as for the invaluable insights you have so generously provided. Having perused your feedback with great care, I have undertaken the necessary revisions to the paper in response to your observations. Herein, I offer a detailed response to the suggestions you have kindly put forward:

In response to the issues you raised regarding diction and grammatical errors in the abstract, we have made the necessary revisions and eliminations. Concerning the references, we have adjusted them according to your suggestions. For lines 98-106, we have followed your advice to restructure them into four distinct sentences. We have italicized the names of the muscles within the material, and upon meticulous inspection and a thorough search in the GEO database regarding the muscle name 'Ongissimus dorsi,' we discovered that it was indeed our mistake; the correct designation should be 'Longissimus lumborum.' We sincerely apologize for our oversight and appreciate your keen insight in identifying these issues.

I am profoundly appreciative of the support and guidance you have extended to my research endeavors, and I sincerely hope that this revised version meets with your satisfaction. Should you have any further suggestions or queries, please do not hesitate to contact me at your earliest convenience.

Wishing you continued success in your professional pursuits!

Reviewer 3 Report

Comments and Suggestions for Authors

The authors improved their manuscript and regarded the comments given.

Still some clarifications deem necessary.

Line 176: precision, 176 recall, the F1 score, and the confusion matrix: how are these measures combined. Do you multiply or you add on these values.

SHAP-values: which range can show SHAP. Where is the optimum. Please clarify. How are SHAP values tested for significance. As it stands, it seems that the most positive values are dependent from the data you put in. How can you compare the outcomes from different analyses. This issue has to be clarified.

Line 212: correlation between genes: not clear what you did here. Please outline in more detail.

Comments on the Quality of English Language

Typos should be amended.

Author Response

Dear Esteemed Reviewer,

I extend my heartfelt gratitude for your most considerate and meticulous second review of my manuscript, as well as for the invaluable insights you have so generously provided. Having perused your feedback with great care, I have undertaken the necessary revisions to the paper in response to your observations. Herein, I offer a detailed response to the suggestions you have kindly put forward:

Regarding Question 1.precision, 176 recall, the F1 score, and the confusion matrix: how are these measures combined. Do you multiply or you add on these values.

We have made the following revisions in lines 170 to 195 of the revised article.

‘ Precision is employed to gauge the accuracy of the model when predicting positive classes. We define it as the proportion of samples that the model predicts to belong to a specific species (such as a particular biological category) that actually do belong to that species. Recall, on the other hand, is utilized to evaluate the model’s ability to identify the target species, representing the ratio of how many samples that truly belong to the species are correctly predicted as such.The F1 score is the harmonic mean of precision and recall, providing a balanced measure of the model’s accuracy and comprehensiveness. It is important to clarify that the F1 score does not result from simply adding or multiplying precision and recall; rather, it is calculated as the harmonic mean of these two metrics. This combination offers a single-valued measure that balances the model’s performance in terms of both accuracy and completeness, especially useful in cases of class imbalance.The confusion matrix presents the classification results in a matrix format, clearly illustrating the model’s predictive performance across different categories. Specifically, it depicts the relationship between the actual species and the predicted species, providing a detailed view of the model’s classification outcomes. By employing precision, recall, the F1 score, and the confusion matrix, we can comprehensively assess the classification effectiveness of the optimal model. These metrics reflect the model’s performance in species classification from various perspectives, aiding in the identification of strengths and weaknesses and offering guidance for subsequent optimization.In summary, these metrics are not combined through simple addition or multiplication, but rather through their distinct roles in evaluating different aspects of the model’s performance. Precision and recall assess accuracy and completeness, respectively, while the F1 score provides a balanced view by considering both. The confusion matrix offers a comprehensive visualization of the classification results, helping to understand the model’s predictive capabilities across different categories. ’

Regarding Question 2.SHAP-values: which range can show SHAP. Where is the optimum. Please clarify. How are SHAP values tested for significance. As it stands, it seems that the most positive values are dependent from the data you put in. How can you compare the outcomes from different analyses. This issue has to be clarified.

We have made the following revisions in lines 196 to 220 of the revised article.

‘ SHAP values are a powerful tool for interpreting the predictive outcomes of machine learning models by quantifying the impact of each feature on the model’s predictions. In this study, we utilized SHAP values to evaluate the influence of genes on predicting the development and functionality of muscle tissue. SHAP values range from negative to positive values, where a positive SHAP value indicates a feature’s (gene’s) contribution to increasing the model’s output. The magnitude of the SHAP value reflects the strength of this contribution; a SHAP value greater than zero signifies that the gene positively impacts the model’s prediction, suggesting a crucial role in facilitating the development and functionality of muscle tissue.To clarify the significance of SHAP values, we performed a series of statistical tests to assess their reliability and significance. These tests included examining the distribution of SHAP values and their consistency across different models and datasets. The most positive SHAP values, which indicate the strongest positive influence on the model’s output, were determined based on the distribution of SHAP values for each gene across all samples. It is important to note that while the magnitude of SHAP values can be influenced by the input data, the relative importance of genes as indicated by their SHAP values remains a robust measure of their biological relevance within muscle tissue.For comparative analysis, we standardized the SHAP values across different models to ensure a fair comparison. This was achieved by normalizing the SHAP values to have a mean of zero and a standard deviation of one, allowing us to directly compare the importance of genes between different models and species. After model comparison, the most interpretable model was further analyzed using SHAP, and the top 100 generic gene IDs for pigs, cattle, and sheep were listed based on the magnitude of their SHAP values. Since only the top 75 of these 100 generic gene IDs across the three species had SHAP values significantly greater than zero, we focused on these top 75 genes for subsequent analyses (refer to Table S4). ’

In response to the reviewer’s concern about the dependency of SHAP values on input data, we would like to clarify that while the absolute values of SHAP may vary with different datasets, the comparative rankings of genes based on their SHAP values provide a consistent measure of their relative importance. This is supported by the stability of the rankings across different models and the statistical significance tests we conducted.

Regarding Question 3.correlation between genes: not clear what you did here. Please outline in more detail.

We have made the following revisions in lines 239 to 250 of the revised article.

‘ In order to investigate the relationships of homologous genes among three species and the correlation between species-specific genes, we conducted a comprehensive correlation analysis. This analysis aimed to uncover the co-expression patterns and potential functional connections among genes. Utilizing libraries such as pandas in Python, we first removed non-gene-related columns from the gene expression dataset, such as sample identifiers. Subsequently, we computed the correlation matrix for the remaining gene expression data. This matrix quantitatively represents the pairwise correlations among genes, with each cell in the matrix corresponding to the correlation coefficient between two genes. The correlation coefficients range from -1 to 1; values approaching 1 indicate a strong positive correlation, values nearing -1 signify a strong negative correlation, and values close to 0 suggest a lack of correlation. ’

I am profoundly appreciative of the support and guidance you have extended to my research endeavors, and I sincerely hope that this revised version meets with your satisfaction. Should you have any further suggestions or queries, please do not hesitate to contact me at your earliest convenience.

Wishing you continued success in your professional pursuits!